# Dataset of Nile Red Fluorescence Readings with Different Yeast Strains, Solvents, and Incubation Times

**Mauricio Ramirez-Castrillon** [1,2,*] **, Victoria P. Jaramillo-Garcia** [2] **, Helio Lopes Barros** [3] **, João A. Pêgas Henriques** [2] **, Valter Stefani** [3,†] **and Patricia Valente** [4]

[1]  Research Group in Mycology (GIM/CICBA), Universidad Santiago de Cali, Calle 5 No. 62-00, Santiago de Cali, Colombia

[2]  Graduate Program in Cell and Molecular Biology, Biotechnology Center, Universidade Federal do Rio Grande do Sul, Av. Bento Gonçalves, 9500 Prédios 43421/43431 - Setor IV - Campus do Vale - CxP. 15005, Porto Alegre, RS CEP 91501-970, Brazil; biovick@gmail.com (V.P.J.-G.); pegas@cbiot.ufrgs.br (J.A.P.H.)

[3]  New Organic Materials and Forensic Chemistry Laboratory (LNMO-QF), Institute of Chemistry, Universidade Federal do Rio Grande do Sul, Av. Bento Gonçalves, 9500, Campus do Vale - CxP. 15005, Porto Alegre, RS CEP 91501-970, Brazil; h.barros@campus.fct.unl.pt

[4]  Department of Microbiology, Immunology and Parasitology, Universidade Federal do Rio Grande do Sul, Rua Sarmento Leite, 500, Porto Alegre, RS CEP 90050-170, Brazil; patricia.valente@ufrgs.br

\*  Correspondence: mauricio.ramirez00@usc.edu.co

†  In Memoriam.

**Abstract:** We used Nile red to estimate lipid content in oleaginous yeasts using a high-throughput approach. We measured the fluorescence intensity of Nile red using different solvents, yeast strains, and incubation times in optimized excitation/emission wavelengths. The data show the relative fluorescence units (RFU) for Nile red excitation, using 1× PBS, 1× PBS and 5% *v/v* isopropyl alcohol, 50% *v/v* glycerol, culture medium A-gly broth, and A-gly broth supplemented with 5% *v/v* DMSO. In addition, we showed the RFU for the Nile red dye for different oleaginous and non-oleaginous yeast strains, such as *Meyerozyma guilliermondii* BI281A, *Yarrowia lipolytica* QU21 and *Saccharomyces cerevisiae* MRC164. Other measurements of lipid accumulation kinetics were shown for the above and additional yeast strains. These datasets provide the guidelines to obtain the optimal solvent system and the minimal interaction time for the Nile red dye to enter in the cells and obtain a stable readout.

**Dataset:** http://dx.doi.org/10.17632/8z22js79dk.1

**Keywords:** oleaginous yeast; lipid; biodiesel; spectrofluorometer

---

## 1. Summary

Nile red (9-diethylamino-5H-benzo[α]phenoxazine-5-one) is one of the most commonly used dyes to quantify neutral lipids in yeasts [1–6]. Different protocols have been proposed to use this dye, but they are not fully standardized, particularly generating some questions about the incubation time (time delay between Nile red contact with cells and fluorescence measurement). For example, a rapid and inexpensive method for the selection of oleaginous strains out of a large collection of yeasts was based on Nile red fluorescence and a microplate reader equipment. In that case, a mixture of DMSO with culture media A (including the cells) was mixed with the Nile red dye, measuring the fluorescence kinetics for 20 min with 60-s intervals to detect the fluorescence peak [2]. Another methodology to

stain yeast cells measured the fluorescence reads immediately after Nile red addition in the solution and recommended good practices in pipetting and mixing the yeast suspension in order to reduce variability in the readout [3]. Later in 2019, a method to stain oleaginous yeasts was proposed using 1× PBS or 20% acetone and proposed a time range of measurements between 5 and 30 min, keeping the samples in darkness [7]. The dataset included in the present work shows the fluorescence readings of Nile red dye using different yeast species, solvents and incubation times. The first part of the dataset reports the fluorescence reads for each variable against the incubation time. The second part shows the fluorescence readings at different incubation times, including several lipid-accumulation kinetics of different oleaginous yeasts. This dataset could be used to predict oleaginous microorganisms using mathematical modeling, to define if the oleaginous ability is species-specific or strain-specific. Additionally, it could help to determine the right incubation time or solvent for each strain, to compare the oleaginous ability of the strains used in this study with other strains.

## 2. Data Description

Table 1 shows all of the yeast strains included in the fluorescence reading dataset. *M. guilliermondii* BI281A [8], *Y. lipolytica* QU21 [9] and *S. cerevisiae* MRC164 [10] were used as the non-oleaginous reference yeast strains. Other strains were also included in the dataset, such as *Papiliotrema flavescens* BI282, BI283, BI296 and *Vishniacozyma* sp. BI237, determining their oleaginous character by gravimetric analysis.

**Table 1.** Yeast strains included in the dataset.

| Strain | Species | Source of Isolation | Oleaginous | Reference |
|---|---|---|---|---|
| **BI281A** | *Meyerozyma guilliermondii* | Flower *Tilandsia gardneri* | Yes | [8] |
| **QU21** | *Yarrowia lipolytica* | Artisanal cheese | Yes | [9] |
| **MRC164** | *Saccharomyces cerevisiae* | Red wine | No | [10] |
| **BI283** | | *Bromelia antiacantha* | Yes | |
| **BI296** | | *Aechmea recurvata* | Yes | |
| **BI282** | | *Bromelia antiacantha* | Yes | |
| **BI091** | *Papiliotrema flavescens* | Flower of *Tilandsia crocata* | ND * | [11] |
| **BI204** | | *Tilandsia gardneri* | ND * | |
| **BI231** | | *Vriesea friburgensis* | No | |
| **BI276** | | *Tilandsia gardneri* | ND * | |
| **BI081** | *Debaryomyces melissophilus* | *Tilandsia gardneri* | No | |
| **BI089** | *Sporidiobolus pararoseus* | Flower of *Tilandsia crocata* | ND * | |
| **BI237** | *VishnIacozyma* sp. | *Vriesea friburgensis* | Yes | |
| **Bel41** | *Occultifur* sp. | | No | |
| **Bel107** | | | ND * | |
| **Bel82** | *Sporidiobolus* sp. | | ND * | |
| **Bel40** | *Candida* sp. | *Bolboschoenus maritimus* | ND * | [12] |
| **Bel106** | *Neophaeomoniella* sp. | | ND * | |
| **Bel49** | Non identified | | ND * | |
| **Bel46** | *Papiliotrema maritimi* sp. nov. | | No | [12,13] |
| **Bel88** | | | No | |

* ND = Non determined, the fluorescence readings showed high relative fluorescence units (RFU) values, but the oleaginous character was not confirmed by gravimetry.

In this dataset, we also assessed the fluorescence reading after different incubation times with Nile red dye diluted in five different solvents, including a mixture of some of them. Figure 1 shows the comparison of fluorescence readings of *Y. lipolytica* QU21 and *S. cerevisiae* MRC164, evaluating several solvents at different incubation times.

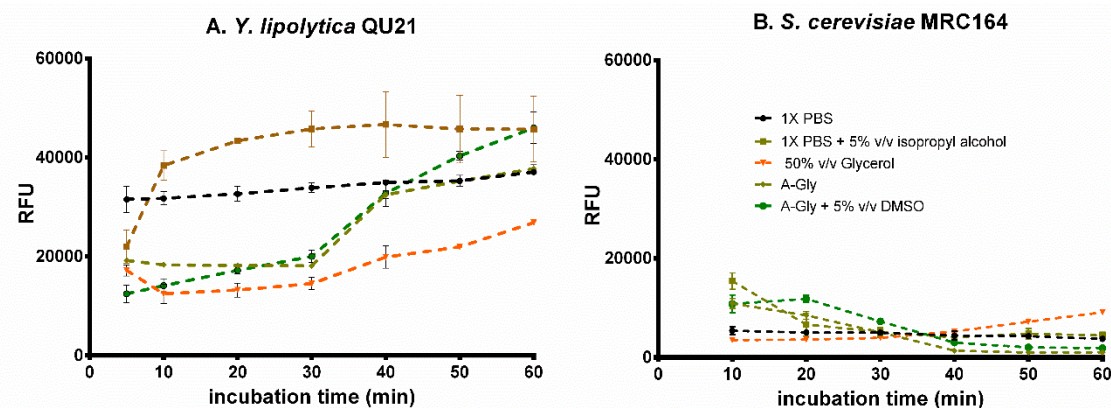

**Figure 1.** Relative fluorescence units (RFU) for *Y. lipolytica* QU21 (**A**) and *S. cerevisiae* MRC164 (**B**), using different solvents against the incubation time. The cell concentration was standardized with an optical density $(OD_{600nm}) = 1$.

Figure 2 shows the fluorescence readings for different solvents and cell densities for *M. guilliermondii* BI281A. The background fluorescence readouts (solvent without cells) are also included.

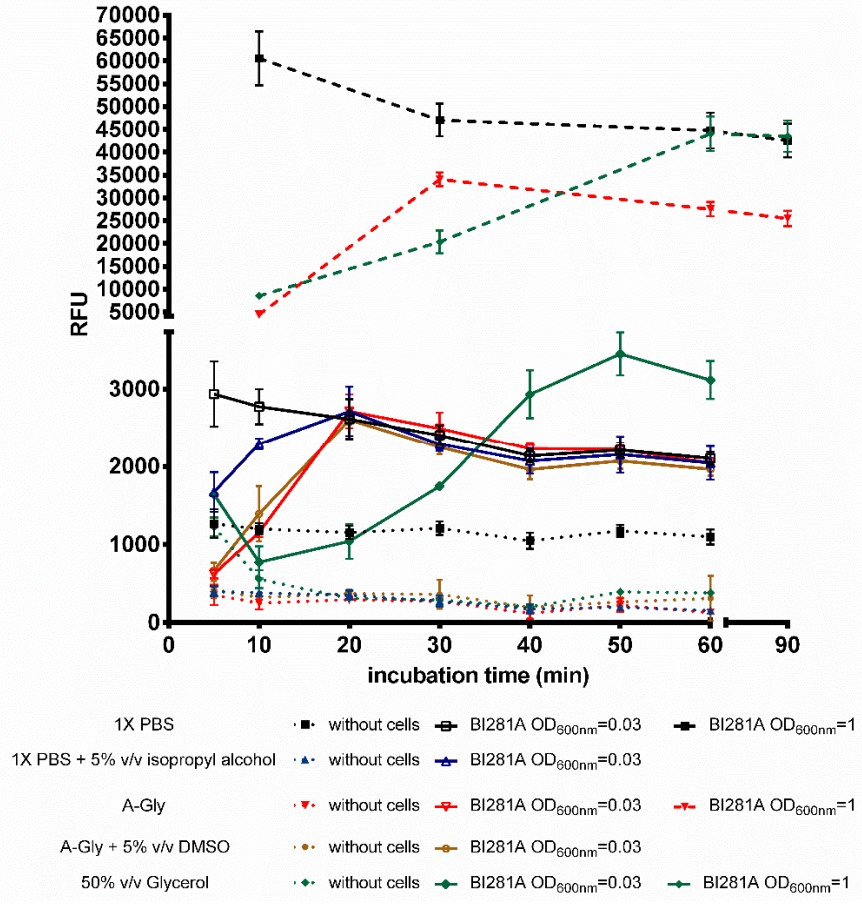

**Figure 2.** Relative fluorescence units for *M. guilliermondii* BI281A using different solvents and incubation time of Nile red. The cell concentration was standardized at optical densities $(OD_{600nm})$ of 0.03 and 1.

Figure 3 shows different kinetics of lipid accumulation for other five yeast strains. For these data, we used A-gly broth as solvent. We measured the fluorescence readings at different times during the yeast growth curve and at different incubation times of Nile red. All the raw data regarding the Figures 1–3 are available in plain text (txt format). The corrected dataset is available in spreadsheet format.

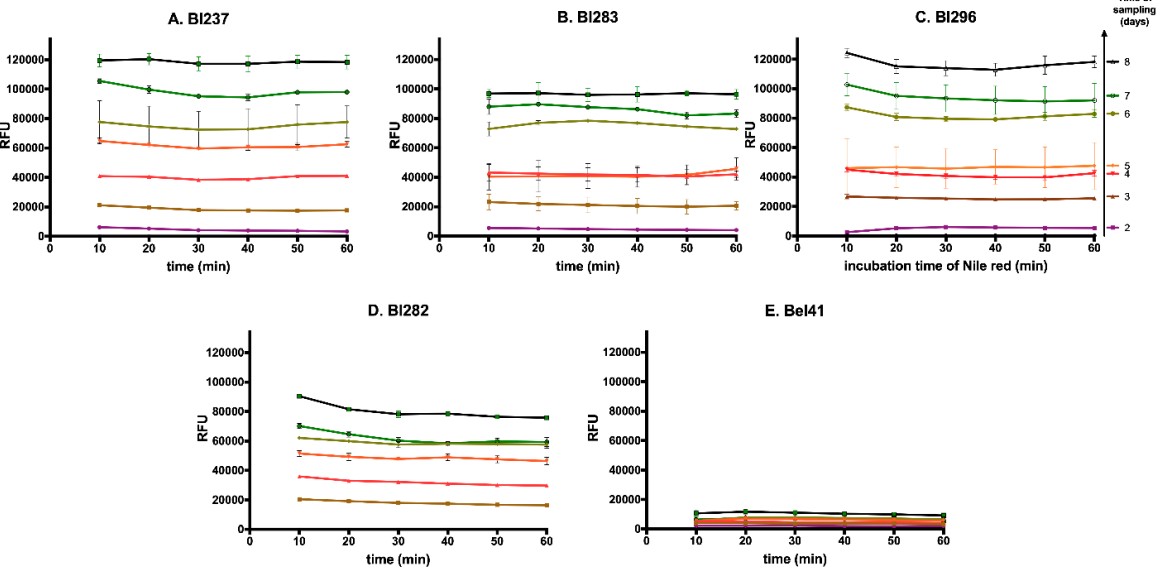

**Figure 3.** Relative fluorescence units for different time of growth for several yeast strains, using A-gly broth as solvent, against the incubation time of Nile red. (**A**) *Vishniacozyma* sp. BI237, (**B**) *P. flavescens* BI283, (**C**) *P. flavescens* BI296, (**D**) *P. flavescens* BI282, (**E**) *Occultifur* sp. Bel41.

We included some secondary data about stability of Nile red in the dataset. Figure 4 shows the consecutive measurement ratio (CMR) for the different fluorescence readings obtained in the raw data. For example, Figure 4A shows the CMR for different solvents in the absence of cells, while Figure 4B,C shows the CMR values for fluorescence readings in presence of *M. guilliermondii* BI281A or *Y. lipolytica* QU21, respectively.

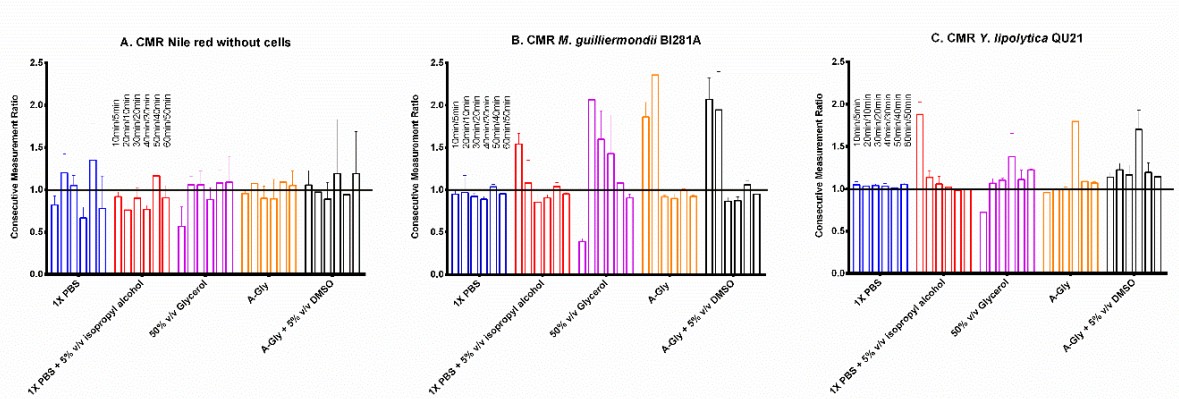

**Figure 4.** Consecutive measurement ratios (CMR) for Nile red. Estimated CMR between times were plotted sequentially for each solvent tested, in the absence of cells (**A**), with cells of *M. guilliermondii* BI281A (**B**) or *Y. lipolytica* QU21 (**C**). The black horizontal line represents a CMR of 1.0. Values are mean ± standard deviation ($n = 3$).

## 3. Methods

The chemical reagents and solvents used were Nile red (Sigma-Aldrich Co., St. Louis, MO, USA) dissolved in acetone (100 µg/mL); 1× PBS (137 mM NaCl, 2.7 mM KCl, 8 mM $Na_2HPO_4$, and 2 mM $KH_2PO_4$); 1× PBS with 5% *v/v* isopropyl alcohol; 50% Glycerol (*v/v* in distilled water); A-gly broth (1 g/L $KH_2PO_4$, 1 g/L $(NH_4)_2SO_4$, 0.5 g/L $MgCl_2$—$6H_2O$ and 15% *v/v* glycerol), and A-gly broth with 5% *v/v* Dimethyl Sulfoxide (DMSO).

The oleaginous yeasts *Meyerozyma guilliermondii* BI281A (deposited as UFMG-CM-Y6124 at Microorganisms Collection, Universidade Federal de Minas Gerais, Brazil) and *Yarrowia lipolytica* QU21 (UFMG-CM-Y327) were tested. *Saccharomyces cerevisiae* MRC164 was used as reference for a non-oleaginous yeast. Each strain was grown in YM broth (3 g/L yeast extract, 3 g/L malt extract, 5 g/L peptone, 10 g/L glucose) for 48 h at 28 °C in order to obtain metabolically active cells. After, each strain was transferred to 75 mL of A-gly broth in a 250 mL flask and grew it for 7, 8 or 12 days at 26 °C and 150 rpm.

The fluorescence readings were obtained with samples containing 150 µL of sample (solvent or cells suspended in each solvent). The solution was transferred to a black background flat bottom 96-well microplates (Jet Biofil, China), and the relative fluorescence was measured in a Perkin Elmer Enspire Multimode Plate Reader 2300 equipment (488 nm of excitation, 585 nm of emission). After measuring the basal fluorescence intensity in each well without the fluorescent dye (background fluorescence), we added 50 µL of Nile Red (final concentration: 25 µg/mL) to the solution, shaken for 5 min inside the equipment and measured the fluorescence in each well with the dye. The measurement was repeated after 5 min, followed by a kinetic reading every 10 min until 60 min. Each measurement was preceded by shaking for 5 s to suspend the cells. We repeated the experiment with different cell densities (for example, $OD_{600nm} = 1$) with measurements at 10, 30, 60 and 90 min of incubation. The fluorescence reading was expressed as RFU (Relative Fluorescence Units), after subtraction of both the background fluorescence of the samples and the fluorescence of the solvent in the presence of Nile red (blank). Each sample had technical triplicates. The ratio of RFU between two consecutive measurements (RFUt + 1 (min)/RFUt (min)) was expressed as the consecutive measurement ratio (CMR), where values closer to 1 indicates low variation between the measurements in times t and t + 1. The gravimetric analysis of biomass and total lipids were performed as described elsewhere [8]. Table 2 shows how data were acquired, the parameters of data collection and data source location.

**Table 2.** Parameters for data acquisition.

| Parameter | Description |
|---|---|
| **Data acquisition** | All data were obtained with a spectrofluorometer (Perkin Elmer Enspire Multimode Plate Reader 2300 equipment). The optical density was measured in a 96-well test plate. The Fluorescence was measured in black background 96-well test plates (Jet Biofil, China). The data was exported by EnSpire Workstation version 3.00. |
| **Data collection parameters** | Volume: Solvent with or without cells: 150 µL; Nile Red (50 mg/L): 50 µL. Final volume: 200 µL.<br>Incubation parameters: 5 min or 10 min with orbital shaking (60 rpm, 3 mm diameter)<br>Fluorescence reading: 488 nm of excitation, 585 nm of emission, 100 flashes by well. |
| **Data source location** | Institution: Universidade Federal do Rio Grande do Sul<br>City/Town/Region: Porto Alegre, Rio Grande do Sul<br>Country: Brazil<br>Latitude and longitude for collected data: 30°04′07.9″ S 51°07′10.9″ W |

**Author Contributions:** Conceptualization, P.V., V.S. and J.A.P.H.; methodology, M.R.-C., V.P.J.-G., H.L.B.; validation, M.R.-C.; data curation, P.V.; writing—original draft preparation, M.R.-C.; writing—review and editing, V.P.J.-G., H.L.B. and P.V.; funding acquisition, P.V. and M.R.-C. All authors have read and agreed to the published version of the manuscript.

**Funding:** This work was funded by Conselho Nacional de Desenvolvimento Científico e Tecnológico (CNPq, Brazil) [Grant numbers 304870/2013-7, 445207/2014-0, 201285/2015-0], Universidad Santiago de Cali (Colombia) [Grant number 934-621119-561], and *Ministerio de Ciencia*, *Tecnologia e Innovación* MINCIENCIAS (Colombia) [Grant numbers 512, 784]. The APC was funded by Universidad Santiago de Cali.

**Acknowledgments:** We thank Luisa M. Nieto Ramirez for her assistance in English manuscript editing.

**Conflicts of Interest:** The authors declare no conflict of interest.

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
