# Peer review of "Dataset of Nile Red Fluorescence Readings with Different Yeast Strains, Solvents, and Incubation Times"

_data, 2020_

Round 1

Reviewer 1 Report

L46-50: I would suggest writing this part using the present verbal form, to clearly indicate that authors are referring to their current work in this part, whereas in the previous part they were referring to previous studies.

The authors should indicate the application of the dataset they are presenting.

Is the dataset reported in the article available? in the manuscript, the authors report "The processed and organized dataset is available in a spreadsheet format.", but they do not indicate where it can be found, and I cannot find supplementary files.

Figure 4: labels are too small to read. I would suggest, instead of writing e.g. "RFU10min/RFU15min", to indicate the timings only, e.g. "10min/15min" and increasing the text size.

Table 1: please indicate what the "ND" (without *) indicates in the table.

Please revise the English. E.g. L127: "Table 2 explain(S) how data were acquired, ...", I would rather use "shows", anyway.

What does "the solvent was fixed as A-gly broth" at line 77 mean?

Author Response

Reviewer 1:

L46-50: I would suggest writing this part using the present verbal form, to clearly indicate that authors are referring to their current work in this part, whereas in the previous part they were referring to previous studies.

R// Corrected.

The authors should indicate the application of the dataset they are presenting.

R// We added a paragraph showing the potential applications of the dataset. “This dataset could be used to predict oleaginous microorganisms using mathematical modeling, to define if the oleaginous ability is species-specific or strain-specific, to determine the right incubation time or solvent for each strain, to compare the oleaginous ability of the strains used in this study with other works.”

Is the dataset reported in the article available? in the manuscript, the authors report "The processed and organized dataset is available in a spreadsheet format.", but they do not indicate where it can be found, and I cannot find supplementary files.

R// Yes, line 29 shows the doi where the dataset was included. http://dx.doi.org/10.17632/8z22js79dk.1

Figure 4: labels are too small to read. I would suggest, instead of writing e.g. "RFU10min/RFU15min", to indicate the timings only, e.g. "10min/15min" and increasing the text size.

R// We replaced figure 4 by another with the suggestion

Table 1: please indicate what the "ND" (without *) indicates in the table.

R// Table 1 has a footnote with the mean of ND. We added the symbol “*” in all the ND abbreviations.

Please revise the English. E.g. L127: "Table 2 explain(S) how data were acquired, ...", I would rather use "shows", anyway.

R// Corrected. We revised the English again with an external scientist. All the modifications are shown in the current version.

What does "the solvent was fixed as A-gly broth" at line 77 mean?

R// Means that A-gly broth was used as a solvent. We modified the text to clarify this paragraph: “For these data, we used A-gly broth as solvent …”

Reviewer 2 Report

I consider manuscript to be of low novelty and interest for the readers, but I leave that for editor´s consideration.

English must be improved throughout the manuscript.

Some criteria should be given to the inclusion of some results in the dataset and not others. For example why data from references [2] and [5] was left out? Since these 2 references are of great and recent impact in the field, having a large number of yeasts screened, authors should assess their dataset.

As an example, I suggest to contact the authors of the two references mentioned above [2] and [5] and, in collaboration with them, to develop some kind of statistical model to predict the oleaginous potential of a certain yeast, using your new database.

In my point of view two important applications can come out of this dataset and this paper in the future.

The first one is to help to clarify the question about lipid content, in yeasts, to be strain dependent and not only species dependent. A recent paper (Miranda et al 2020, "Modified high-throughput Nile red fluorescence assay for the rapid screening of oleaginous yeasts using acetic acid as carbon source") - the authors proved that lipid content varied greatly with the yeasts´ strain, despite being in the past associated only to the species level. With the current manuscript´s dataset one can understand if the differences found when doing new experiments are due to variation between strains (and between individuals) or are due to the experimental variation as discussed in the work (variation according to experimental parameters).

The second point is related with new experiments that can arise after the work is published. The authors are encoraged to add some ideas on potential mathematical approaches that could be applied to the dataset in order to analyse and predict which are the parameters that most influence lipid content in yeasts.

Author Response

Reviewer 2:

English must be improved throughout the manuscript.

R// We revised the English again with an external scientist. All the modifications are shown in the current version.

Some criteria should be given to the inclusion of some results in the dataset and not others. For example why data from references [2] and [5] was left out? Since these 2 references are of great and recent impact in the field, having a large number of yeasts screened, authors should assess their dataset.

As an example, I suggest to contact the authors of the two references mentioned above [2] and [5] and, in collaboration with them, to develop some kind of statistical model to predict the oleaginous potential of a certain yeast, using your new database.

R// We included in this data description only our dataset. When we revised the reference [2]( Sitepu et al. 2014) is already a review with high impact. The reference [5] (Miranda et al. 2020), like many others that we cited, assessed a large collection of yeasts to find new oleaginous strains. We don´t have these datasets, therefore we are not able to include them in our data description. However, we thank the suggestion to contact the authors to collaborate with them.

In my point of view two important applications can come out of this dataset and this paper in the future.

The first one is to help to clarify the question about lipid content, in yeasts, to be strain-dependent and not only species-dependent. A recent paper (Miranda et al 2020, "Modified high-throughput Nile red fluorescence assay for the rapid screening of oleaginous yeasts using acetic acid as carbon source") - the authors proved that lipid content varied greatly with the yeasts´ strain, despite being in the past associated only to the species level. With the current manuscript´s dataset one can understand if the differences found when doing new experiments are due to variation between strains (and between individuals) or are due to the experimental variation as discussed in the work (variation according to experimental parameters).

R// Thanks for the suggestions. Since the review published by Sitepu et al. (2014) the hypothesis that the oleaginous ability is strain-dependent, and not species-dependent, has been supported by several reports. For example, references [5], [8] and several strains of Papiliotrema flavescens (included in this dataset) supported this hypothesis by gravimetric measurements, but up to now it was not supported by fluorescence readings. For this reason, we included in Table 1 a column extrapolating the results if the experimental procedure was done with the strains showed in the dataset. We showed that the experimental variation due to different incubation times or solvents could be limiting in these comparisons with fluorescence readings to show new oleaginous strains.

The second point is related with new experiments that can arise after the work is published. The authors are encoraged to add some ideas on potential mathematical approaches that could be applied to the dataset in order to analyse and predict which are the parameters that most influence lipid content in yeasts.

R// Thanks again, we are not experts in mathematical modeling, but we added a paragraph in the text to invite experts in this area to work with our dataset. "This dataset could be used to predict oleaginous microorganisms using mathematical modeling, to define if the oleaginous ability is species-specific or strain-specific, to determine the right incubation time or solvent for each strain, and to compare the oleaginous ability of the strains used in this study with other works."